# Exploring Fungal Communication Mechanisms in the Rhizosphere Microbiome for a Sustainable Green Agriculture

**DOI:** 10.3390/jof11100726

**Published:** 2025-10-09

**Authors:** Jing Gao, Anqi Dong, Jiayi Li, Jiayu Xu, Zhihong Liang, Antonio Francesco Logrieco

**Affiliations:** 1Xianghu Laboratory, Zhejiang Provincial Laboratory of Agriculture, Hangzhou 311231, China; gjing1997@126.com (J.G.); xjiayu1999@126.com (J.X.); 2College of Food Science and Nutritional Engineering, China Agricultural University, Beijing 100083, China; m19838924611@163.com (A.D.); lijiayi@nwafu.edu.cn (J.L.)

**Keywords:** rhizosphere microbiome, quorum-sensing molecules, biofilms, compound microbial agents, biological antagonism

## Abstract

In the long-term evolutionary process, species maintain a natural balance within certain limits through communication. As plants grow and function as producers, root enrichment fosters a dynamic rhizosphere microbiome, which serves not only as a disintegrator within the ecological niche but also as a medium for interaction between the host and the soil environment. The life cycle of fungi within the microbiome alternates between single-cell resting spores and multicellular trophic mycelia. This cycle not only establishes a stable rhizosphere environment but also plays a crucial role in regulating both intra- and interspecific information transmission, significantly impacting the environment and plant health. The rhizosphere microbiome, particularly the fungi it contains, can be harnessed to repair environmental damage and either promote the growth of the plant host or inhibit pathogens. However, the mechanisms underlying these actions remain inadequately understood, hindering the advancement of artificial regulation. Additionally, the variability of influencing factors, along with unstable genes and traits, poses challenges to industrial development. In conclusion, this paper focuses on the fungal components of the rhizosphere microbiome, introduces the mechanisms of communication and current applications, and further analyzes existing bottlenecks and potential solutions. The aim is to provide theoretical support for achieving green, sustainable agriculture through biological means.

## 1. Introduction

In natural ecosystems, species populations exhibit characteristic dynamics (e.g., regulated growth, niche differentiation) that arise from biotic interdependence (e.g., symbiosis, competition) as well as constraints imposed by abiotic factors (e.g., nutrient availability, temperature) and stochastic processes, collectively shaping balanced ecological networks. Microorganisms serve as primary producers and decomposers, playing a pivotal role in sustaining material circulation, energy flow, and ecosystem stability within natural systems. To secure competitive ecological niches, where microorganisms gain advantages in resource acquisition or functional expression, individual microorganisms engage in essential cellular activities and demonstrate acclimatory stress responses to cope with environmental challenges (e.g., nutrient scarcity, fluctuating pH, or oxidative stress). These stress-induced physiological adjustments often involve the secretion of diffusible signaling molecules (DSMs) or metabolites, which simultaneously mediate intraspecific coordination and act as cues to recruit compatible partners [1]. Soil, as a three-phase polydisperse medium (solid, liquid, gaseous), harbors extraordinarily diverse biological resources, including unique microbial communities and specialized invertebrates. While some taxa (e.g., certain protozoa) overlap with aquatic environments, soil’s distinct microhabitats (e.g., rhizospheres, soil pores) support high levels of taxonomic and functional uniqueness that differ from the free-living communities of air and bulk water [2]. The rhizosphere, which defined as the soil microdomain modified by plant root activities to possess distinct physical, chemical, and biological properties relative to bulk soil, is considered one of the most dynamic interfaces on Earth [3]. The microorganisms associated with the surrounding rhizosphere are collectively known as the “rhizosphere microbiome” and form a network of interactions with plant hosts and other soil-dwelling organisms across different trophic levels [4]. Analogous to the current hot point “gut microbiota”, the rhizosphere microbial community associated with the plant host constitutes an integrated system, which was discovered earlier and should receive more attention [5]. Through the plant’s structure, they facilitate the exchange of matter and energy, collectively encoding a greater number of genes than the host itself, a phenomenon termed the plant’s “second genome” [6].

The rhizosphere microbiome comprises a complex assemblage of plant roots, along with bacteria, fungi, archaea, algae, actinomycetes, protozoa, and viruses, exhibiting a total cell density ranging from 10^6^ to 10^9^ cells/cm^2^ [7]. Fungi constitute a significant portion of the rhizosphere microbial biomass and are instrumental in promoting plant growth and ameliorating soil quality due to their unique growth and metabolic patterns [8]. However, infections caused by toxigenic fungi in crops can lead to crop products contamination by mycotoxins. This contamination initiates during cultivation, whereby dormant spores attached to the surface or vegetative mycelia invade the plant. Upon detachment from the soil microenvironment, the ecosystem balance is disrupted, triggering the rapid activation of defensive mechanisms in toxigenic fungi, such as the upregulation of secondary metabolism, which results in the production of mycotoxins throughout the subsequent food chain, including collection, storage, transportation, and marketing. According to the Food and Agriculture Organization (FAO), annual mycotoxin contamination accounts for 25% of the global grain supply and has been reported in over 100 types of agricultural products, including food and feed, with oil crops being particularly susceptible [9]. Recently, the prevalence and distribution of mycotoxins have escalated due to the compounded effects of extreme climatic conditions and emerging pollutants [10].

In addition to fluctuations in temperature, humidity, pH, and other environmental factors that contribute to abnormal mycotoxin production and variability, disruption of the “soil–plant–fungus” equilibrium may also induce behavioral transformations in toxigenic fungi within the rhizosphere microbiome [3]. Communication signaling constitutes a critical facet of the adaptive resistance mechanisms employed by fungi. Furthermore, how does intraspecific communication, represented by quorum sensing, respond to influences from plant roots and other rhizosphere microorganisms? These emerging research directions remain active areas of investigation.

Additionally, the controlled regulation of microorganisms to safeguard host organisms, mitigate losses, promote growth, and enhance economic returns has emerged as a prominent focus in both agricultural and forestry sciences. Nature harbors immense biological resources, and the utilization of microbial agents and cell factories has shown remarkable potential. However, many of these effects remain inconsistent. The intricate biological interactions arising from ecological dynamics, along with incompletely elucidated internal mechanisms, constrain the advancement of rhizosphere microbiology research.

This review focuses on fungi within the rhizosphere microbiome, elaborating on the principles governing their assembly and establishment in response to dynamic host changes, as well as the current state of research in this field. It analyzes key regulatory aspects and proposes potential mechanisms that refine knowledge of microbial cell communication, findings intended to further propel the development of sustainable green agriculture.

## 2. Methodology

To ensure the review’s comprehensiveness, we conducted a systematic literature search across three core databases—Web of Science, Scopus, PubMed—with a timeframe focused on recent advances. We used tailored keyword combinations that integrated terms for the biological system (“rhizospheric fungi,” “rhizobiome”), target molecules (“diffusible signaling molecules,” “quorum sensing”), and ecological context (“plant-microbe interaction”), ensuring that we captured both foundational and cutting-edge research.

The study of rhizosphere interactions involves the intersection of a series of disciplines such as biochemistry, microbiology, bioinformatics, and genetics. Core methodological approaches include but are not limited to gas/liquid chromatography–mass spectrometry, biosensors, gene editing/sequencing, fluorescence tracking, molecular simulation, etc., in the research of rhizosphere interactions (see Table 1).

## 3. Assembly of Fungal Communities in the Rhizosphere Microbiome and Their Species Composition

### 3.1. Dynamic Building with Soil, Plant, and Fungi

The interrelationship among host roots, their associated microbiome, and the surrounding soil environment forms a cohesive system that engages in dynamic interactions to maintain ecological balance amid environmental fluctuations [11]. The biological activities and species characteristics within this system exhibit significant variability, reflecting the extensive influence of environmental factors. Furthermore, the adaptive feedback mechanisms of the rhizosphere microbiome in response to environmental changes help sustain the balance of this system [12,13]. This microbiome acts as a crucial regulator of material transformation during ecosystem evolution (see Figure 1).

The macro environment serves as a fundamental selective pressure for microorganisms, which in turn can influence soil properties [14]. Soil texture affects aeration, water retention, and nutrient content, thereby impacting the living environment. Additionally, soil composition influences microbial growth and metabolism, ultimately determining the assembly of the microbiome in a given environment [15]. Microorganisms can enhance soil aggregate structure by secreting extracellular polymers (EPS), which increase both stability and aeration [16]. Certain microbial strains contribute to soil nutrient cycling, including nitrogen fixation, organic matter decomposition, and the mineralization of organic carbon [17]. Furthermore, the metabolic activities of specific strains can alter soil pH, while some microorganisms possess the ability to degrade organic pollutants or immobilize heavy metals, thereby improving environmental quality [18].

Planting further enriches the diversity of microorganisms. Various host-related factors, including developmental stage and circadian rhythms, influence the structure of the root microbial community [19]. This structure is also partially determined by uncontrollable abiotic factors in the natural environment, which elucidates the challenges associated with culturing many environmental microorganisms, particularly endophytes [20]. Physically, the soil’s water release characteristics and aeration conditions in the rhizosphere are superior to those in non-rhizosphere areas due to the interpenetration of plant roots. Additionally, the diversity of root-associated microorganisms tends to decrease from the surrounding soil to the rhizosphere, root surface, and root interior, indicating a high degree of specialization [21]. Chemically, approximately 20% to 60% of photosynthates produced during plant growth are released into the rhizosphere as root exudates [22]. These exudates can be categorized into two primary categories: small molecular compounds (including amino acids, organic acids, sugars, alcohols, polyamines, fatty acids, purines, plant hormones, terpenes, flavonoids, benzoxazines, etc.) and macromolecular compounds (including polysaccharides and proteins). Root exudates serve multiple functions: (A) surfactants, such as lecithin, help create a hydrophobic environment; (B) organic matter, along with apoptotic roots and their exfoliations—including root hairs, epidermal cells, and root crowns—contributes to a nutrient-rich root environment, serving as vital sources of nutrients and energy; (C) small root exudates function as signaling molecules, with attractants and repellents facilitating the recruitment of rhizosphere microorganisms [23,24]. Particularly, quorum-sensing molecules (QSMs), including volatile compounds, regulate interactions among plants and microorganisms, as well as among microorganisms themselves [25].

The types and concentrations of root exudates vary across different root regions, resulting in distinct microbial niches on a spatial scale [26]. The absolute abundance of microorganisms is highest in the rhizosphere, followed by the root surface, within the roots, and lowest in the surrounding soil [27]. Additionally, characteristics such as motility, chemotaxis, and adhesion influence the chemotactic behavior of microorganisms as they assemble toward the root exudates of the plant host. This spatial heterogeneity allows rhizosphere microorganisms to exert varying effects on plants across various vertical and horizontal scales [28]. Based on absolute abundance to elucidate the assembly processes of the rhizosphere microbiome, Microorganisms are initially amplified in the rhizosphere soil and subsequently selected through interactions with the root, leading to the formation of a specific rhizosphere microbial community [29]. Furthermore, there are also “two-step selection” and “multi-step enrichment” models, which emphasize the gradual selection of microbial communities from the soil to the rhizosphere and ultimately into the roots [27,30]. Although the rhizosphere microbiome often fluctuates with changes in the soil environment, host genetics acts as a ‘selection filter’ that maintains consistent assembly of a core microbiome subset within a particular host genotype, regardless of soil conditions. Moreover, the rhizosphere microbiome undergoes a stabilization and host-specialization dynamic throughout plant growth: it shows high compositional variation in the early stages and becomes progressively less variable while more host-specific in the later stages [8].

### 3.2. Species and Functions of Rhizosphere Fungi

Rhizosphere fungi encompass a phylogenetically and functionally diverse array of taxa, including symbionts that form mutualistic associations with plants, saprotrophs that drive organic matter cycling, regulators that modulate microbiome structure, and pathogens that occasionally disrupt plant health. These groups collectively shape rhizosphere function and plant fitness, with each contributing unique ecological roles [31]. Among the microbial inhabitants of the rhizosphere, fungi can establish a more reciprocal symbiotic relationship with higher plants compared to bacteria. They develop extensive rhizosphere mycelium through mycelial extension in the soil, which is influenced by mycelial secretions, occupying an irreplaceable ecological niche within the ecosystem [32]. The interactions between rhizosphere fungi and their plant hosts can be categorized into three types: neutral commensal, harmful pathogenic, and beneficial promoting relationships. The majority of rhizosphere fungi engage in neutral symbiosis with plants, demonstrating a relatively loose dependence while also being attracted to plant secretions. Although their influence may not be overtly positive or negative, under certain conditions, neutral fungi can exhibit either beneficial or detrimental effects on their host [33].

Beneficial rhizosphere fungi can influence their host through bidirectional regulatory mechanisms. Nutrient exchange and symbiotic formation are facilitated by signaling molecules secreted by plants, which stimulate fungal mycelia to accelerate growth and branching, thereby enabling them to approach and invade host roots [6]. This interaction expands the plant’s absorption area and enhances the utilization of nutrients and mineral elements. For example, arbuscular mycorrhizal fungi (AMF) are attracted by plant-secreted strigolactones, which prompt the fungi to bifurcate and form unique arbuscular structures that can supply over 70% of the total phosphorus acquired by the host plant [34]. Concurrently, fungi can dynamically regulate the balance of nutrient exchange through feedback regulatory pathways that prevent energy wastage and excessive reliance on the host. Furthermore, they are influenced by hormones and secondary metabolites [35]. AMF can secrete various hormones, such as abscisic acid [36], auxin [37], and vincristine, which promote root growth in plants and enhance their stress resistance. Additionally, fungi inhibit pathogens to protect their hosts. Rhizosphere fungi can cooperate with and promote the growth of beneficial microorganisms, such as azotobacter, phosphorolytic bacteria, and actinomycetes, thereby occupying a dominant ecological niche that mitigates pathogen colonization in the rhizosphere [38]. Moreover, their synergistic interactions regulate host root exudates to construct a defensive barrier that directly repels pathogen invasion. Plants inoculated with AMF can enhance resistance genes’ expression and elevate phenolic acids, flavonoids, enzymes and other anti-pathogen compounds, thereby alleviating pathogen infections [39]. Finally, specific metabolites produced by certain rhizosphere fungi can modify environmental conditions in the rhizosphere, such as pH and redox potential, which can inhibit pathogen survival.

The detrimental effects of fungi on their hosts encompass both direct disease and indirect inhibition; what is particularly concerning is that much of the damage occurs simultaneously. Mycelial infections of roots can cause physical damage and potentially lead to plant mortality, while also diminishing the host’s stress resistance through the production of toxic metabolites, thereby impeding normal growth and development. Common pathogenic and rot fungi include *Fusarium*, *Alternaria*, *Verticillium*, and *Pythium*, which can induce conditions such as root rot, wilt, and hollow disease (see Table 2). Although non-pathogenic fungi do not directly cause plant diseases, they can still be harmful by inhibiting beneficial rhizosphere microorganisms from competing with plants for nutrients, thereby indirectly reducing biomass. Additionally, these fungi may pose threats through mechanisms such as spore adhesion, rapid growth, and the metabolic accumulation of mycotoxins once they exit the soil environment.

Recently, a novel classification system for the rhizosphere microbiome based on distinct assembly mechanisms be published, which delineates the microbiome into two dominant factors: environmental dominance, which is influenced by rhizosphere soil characteristics (approximately 96.5%), and plant genetic dominance, which is determined by the plant host genotype (less than 3.5%) [61]. This classification framework enhances our understanding of the assembly mechanisms of the rhizosphere microbiome and provides a new perspective for future research endeavors.

## 4. Inter- and Intra-Specific Communication of Rhizosphere Fungi

The rhizosphere microbiome represents a distinct microenvironment characterized by a diverse array of organisms and abiotic components within a specific spatiotemporal context. This microbiome constitutes a complex network of interrelated factors that sustaining a dynamic equilibrium and exhibit a certain degree of buffering capacity as a cohesive entity. Within the rhizosphere microbiome, various microbial species, along with fungi and plants, engage in interactions through direct contact facilitated by structures such as vesicles and mycelia, as well as through non-contact intercellular secretory systems that utilize diffusible molecules. Although multiple mechanisms underpin these interactions, our understanding of the precise processes involved remains insufficiently developed (see Figure 2).

### 4.1. Hyphosphere

The hyphosphere is a specialized zone within the soil, characterized by the proliferation of fungal mycelia, which distinguishes it from typical soil environments. This zone closely overlaps with plant roots, forming the rhizosphere mycelium. During dynamic interactions with plants, specific microbial communities are recruited and ultimately develop into the rhizosphere microbiome. The mycelial network established by mycelia maintains the stability of this microenvironment and constitutes a diffusion network for signaling molecules, thereby expanding the sensing range of microbial interactions. This phenomenon is commonly referred to as the “fungal superhighway”, with typical filamentous fungi such as *Aspergillus* and *Mucor* play an important role [62,63].

In terms of resource allocation, the hyphosphere regulates the distribution of resources within the soil to mitigate competition among plants and maintain ecological balance. Organisms within the hyphosphere engage in cooperative and symbiotic relationships to compensate for the fungi’s limitations in utilizing organic nutrients, primarily by producing enzymes and facilitating the mineralization of organic matter [17]. Concurrently, the mycelium in the rhizosphere enhances the host plant’s nutrient absorption capabilities and facilitates the transfer of energy within the plant system [38]. Regarding interspecific communication, mycelium serves as a mediator for the transmission of information both within individual plants and among different species. The mycelial network in the soil can convey defense signals secreted by the host in response to pest attacks to neighboring plants, thereby assisting them in addressing potential threats [64]. For instance, the spores of non-spore-forming *Streptomyces* can activate the chemotactic mechanisms of soil bacteria such as *Bacillus subtilis* and *Pseudomonas fluorescens* [65], prompting their migration from the bulk soil to the nodule layer of plant roots [66]. This interaction fosters a more robust legume-rhizobia symbiosis, with *Streptomyces* playing a pivotal role in this ecological relationship.

### 4.2. Diffusible Signaling Molecule

Diffusible signaling molecules (DSMs) are synthesized within cells and subsequently secreted into the extracellular environment by both plant and microbial organisms. These molecules facilitate dynamic interactions in a non-contact manner, thereby expanding their influence [67]. A diverse array of signaling molecules exists, and even identical molecules can elicit markedly different responses, potentially attributable to variations in the signaling detection and response mechanisms across different species [68] (see Table 3).

The significance of plant-derived signaling molecules in mediating interactions between plants and rhizosphere fungi is noteworthy. Throughout their growth, plants generate a variety of molecules that orchestrate their physiological processes at various developmental stages [69]. For example, hormones such as indole-3-acetic acid (IAA), cytokinins (CTK), gibberellins (GA), ethylene (ET), and abscisic acid (ABA) play crucial roles in organ development [70]. Furthermore, in response to environmental stressors, plants emit volatile compounds such as nitric oxide (NO) that serve as signals to neighboring plants, thereby enhancing the collective responsiveness of plant populations [71]. Additionally, signaling molecules produced by plants not only function within the plant system but are also secreted into the rhizosphere to modulate the activity of microorganisms [72]. Conversely, microorganisms can synthesize and secrete analogs of plant signals that penetrate the host from the rhizosphere, thereby extending their influence to the aerial parts of the plant. For instance, the compound 3-oxo-C12-HSL, produced by *Pseudomonas aeruginosa*, can readily infiltrate plant tissues and impact their nutritional and defense mechanisms [73].

DSMs facilitate both intra- and interspecific communication among microbial populations. The rhizosphere is inhabited by over 30,000 microbial species, with cell densities ranging from 10^10^ to 10^12^ cells per gram of soil—approximately 1000 to 2000 times greater than those found in non-rhizosphere soils [27]. This elevated population density not only enhances the likelihood of cellular communication but also intensifies competition for survival resources among microorganisms. In response to the environmental pressures associated with high cell density, microorganisms employ quorum-sensing (QS) mechanisms to stabilize the spatial distribution of the rhizosphere microbiome [74]. Quorum-sensing molecules (QSMs) are synthesized intracellularly and subsequently secreted extracellularly, mediating adaptive behavioral changes in microorganisms [75]. While quorum sensing is prevalent in both bacterial and fungal species, the potential for QSMs to mediate cross-species regulation remains uncertain. Bacterial QSMs can be classified into N-acyl-homoserine lactones (AHLs) in Gram-negative bacteria, autoinducing peptides (AIPs) in Gram-positive bacteria, and autoinducer 2 (AI-2), which functions as a universal signaling molecule among both Gram-positive and Gram-negative bacteria [76]. In fungi, QSMs encompass alcohols, oxylipins, small peptides, aldehydes, and volatile organic compounds (VOCs) [75,77]. Beyond their role as communication molecules that can globally regulate transcriptional activity among microbial cells, QSMs can also exert direct influences on plant hosts, even in the absence of quorum-sensing mechanisms within the plant [64]. Once the concentration of QSMs within the rhizosphere microbiome surpasses a certain threshold, they can synchronize microbial community behavior and indirectly affect the host organism. At present, the types that can be used both as fungal QSMs and biocontrol products include peptides and aldehydes; however, more specific compounds have yet to be refined, indicating potential for further development.

**Table 3 jof-11-00726-t003:** Diffusible Signaling Molecules (DSMs) Produced by Rhizospheric Fungi and Their Communication Roles.

Fungi	Produced Molecules	Communication Type (Intraspecific/Interspecific/Interkingdom)	Function	References
*Candida* spp.	Farnesol Tyrosol γ-butyrolactone Phenylethanol, 3,4-Dihydroxyphenyl ethanol	Intraspecific	Regulate morphological transitions (yeast-to-hyphae); control biofilm maturation and quorum sensing; coordinate virulence gene expression.	[78,79]
*Aspergillus* spp.	Gliotoxin Pyomelanin 1-Octen-3-olCyclopiazonic acid	Intraspecific & Interspecific (bacteria/fungi)	Mediate competition in microbial communities via antimicrobial activity; regulate fungal sporulation.	[80,81]
*Trichoderma* spp.	6-Pentyl-α-pyrone (6-PAP) Trichovirin Volatile terpenes Trichodermin Chitinase-inducing factors	Interspecific & Interkingdom (Plant)	Inhibit phytopathogens (e.g., *Fusarium oxysporum*); induce plant systemic resistance via jasmonic acid signaling.	[82]
*Penicillium* spp.	Penicillic acid Sesquiterpenoids Roquefortine C PR-toxin Citrinin	Intraspecific & Interspecific	Regulate fungal growth; modulate plant defense responses in fruits (e.g., apples).	[83]
*Rhizopus* spp.	Putrescine Cadaverine Spermidine Ethylene precursor (1-aminocyclopropane -1- carboxylic acid)	Intraspecific & Interkingdom (Plant)	Promote fungal spore germination; enhance plant root elongation and nutrient uptake.	[84]
Arbuscular Mycorrhizal Fungi (AMF) e.g., *Glomus* spp.	Strigolactone analogs Flavonoid-inducing factors Mycorrhizal lipochitooligosaccharides (Myc-LCOs) Strigolactone mimics	Interkingdom (Plant–Fungus)	Trigger fungal spore germination and hyphal branching, stimulate plant root colonization and phosphate transport; facilitating the establishment of mycorrhizal symbiotic relationships.	[85]
*Saccharomyces cerevisiae*	2-Phenylethanol Maltol Isoamyl alcohol 2-Methylpropanol	Intraspecific	Regulates yeast flocculation and fermentation processes; participate in signal transmission for nutrient exchange during symbiosis with plants; inhibit bacterial biofilm formation (e.g., *E. coli*).	[86,87]
*Fusarium* spp.	Fusaric acid Fumonisins Beauvericin Zearalenone	Interspecific & Interkingdom (Plant)	Suppress plant immune responses; modulate plant hormone signaling (auxin pathways); facilitate fungal invasion of cereal crops (e.g., wheat).	[88]
*Metarhizium* spp.	Volatile organic compounds (e.g., 1-octen-3-ol, 3-Octanone) Destruxins	Interkingdom (plants/insects)	Stimulate plant growth via auxin-like activity; attract insect hosts for fungal pathogenesis.	[89]
*Piriformospora indica*	Indole-3-acetic acid (IAA) Gibberellin A3 Sphingolipids	Interkingdom (plants)	Promote plant growth through hormone signaling; enhance stress tolerance.	[90]

### 4.3. Biofilm

Biofilm formation is a significant phenomenon associated with quorum sensing. As the density of microorganisms in a specific area increases, the concentration of secreted quorum-sensing molecules reaches a threshold that triggers the aggregation of free microorganisms into a multicellular biofilm that adheres to a host surface [91]. It is estimated that between 40% and 80% of microorganisms on Earth exist in biofilm form, which enhances communication and cooperation among these organisms, thereby enabling them to exploit ecological niches more effectively [92].

Alpkvist has proposed a three-dimensional model of biofilm architecture, which consists of an inner structure comprising the basal layer, conditional layer, and connection layer, arranged from the innermost to the outermost layer, with the outermost part enveloped by the biofilm layer [93]. Extracellular polymeric substances (EPS) secreted by microorganisms, including polysaccharides, proteins, nucleic acids, and lipids, account for over 90% of the biofilm’s composition [94]. Various protein components, particularly matrix proteins, play a crucial role in promoting the formation, development, and maturation of biofilms. Additionally, extracellular DNA (eDNA) and lipids present on the biofilm surface can interact with positively charged elements within the matrix, thereby contributing to the stability of the biofilm structure [95].

Biofilms that are firmly attached to plant roots create a relatively independent and enclosed environment, which enhances the colonization potential of internal microorganisms and strengthens the interactions among microbes, plants, and soil [96]. Additionally, the interior of the biofilm encapsulates and stabilizes a greater quantity of nutrients, thereby improving resource utilization by microorganisms and enhancing their resilience to various environmental stresses, including fluctuations in temperature, osmotic pressure, and pH levels. Biofilms also serve as significant mechanisms for virulence and resistance [97]. During biofilm formation, microorganisms can initiate a series of defensive responses, including the synthesis of antibiotics, degrading enzymes, and other antibacterial substances, as well as inducing resistance to antimicrobials and evading the host immune system. This enhanced capability also facilitates more stable horizontal gene transfer through the exchange of extracellular genetic material within the biofilm [98].

## 5. Application and Development

The rhizosphere microbiome is a crucial component and medium for plant interactions with soil and the atmosphere. It is characterized by its dynamic nature and its ability to sustain the stability of both the plant host and the soil environment. Notably, this multifunctional assemblage comprises microorganisms that exhibit strong self-healing properties and considerable potential for artificial manipulation. Consequently, rhizosphere microorganisms are expected to play an increasingly important role in agricultural production, environmental protection, biotechnology, and various other fields. However, the lack of clarity regarding the regulatory mechanisms governing individual microorganisms, as well as their adaptive strategies in response to environmental uncertainties, presents challenges to their development. Therefore, research and applications concerning rhizosphere microorganisms are focused on advancing more precise and efficient methodologies.

### 5.1. Ecological Restoration

Characteristic species or communities within the rhizosphere microbiome have been employed as indicators of overall soil health. This approach facilitates multifaceted monitoring of various soil types, including sandy soils, those with acidic or alkaline properties, and soils differing in metal content and organic pollutants [99]. Rhizosphere microorganisms are essential for the healthy growth of their host plants, as they possess sensitive defense mechanisms against environmental stressors [100]. For example, they produce EPS with diverse types and structures that help retain water, enhance the absorption of metal ions and organic matter, and play a significant role in the remediation of contaminated soil and water (see Table 4).

Exopolysaccharides secreted by rhizosphere microorganisms constitute over 90% of biofilm composition, which includes polysaccharides, proteins, nucleic acids, and lipids. These exopolysaccharides exhibit significant biosorption and flocculation properties, including scleroglucan, beta-glucan, split-fold polysaccharide, pullulan, and galactan, among others [101]. Investigations have identified genes associated with polysaccharide synthesis in biofilms, ranging from 12 to 17 kilobases in length, which primarily participate in the processes of monosaccharide addition, acylation of monosaccharides, monosaccharide polymerization, and polysaccharide secretion [102]. The anionic groups of polysaccharides, such as hydroxyl, carboxyl, amino, and sulfate, are rich in coordination atoms like oxygen and nitrogen, which can form complexes with ions. Additionally, these groups can interact with metal ions and organic matter, encapsulating them through alterations in membrane structure, thereby facilitating the degradation and metabolism of pollutants through the synergistic actions of biofilm constituents [103].

**Table 4 jof-11-00726-t004:** Fungi for ecological restoration.

Type	Contaminant	Application	Repair	Species	References
Cellulose decomposing fungi	micropollutants	Forest and grassland	Secreting cellulase and lignin enzyme to degrade large plants and promote ecological cycle	*Phanaerochaete chrysosporium* *Pleurotus ostreatus*	[104]
arbuscular mycorrhizal fungi	Barren land	Sandy land	Form symbiotic relationship with plant roots to enhance plant resistance to abiotic stresses, thus improving soil structure	*Rhizophagus Diversispora epigaea* *Gigaspora rosea*	[105]
Halophilic fungi	Rigid, deteriorating soil	Mining areas Saline-alkali land	Adapt and improve soil structure and fertility through metabolism to promote vegetation recovery	* Aspergillus glaucus *	[106]
Plastic degrading fungi	Plastic polymer	litter-piled field	Biological reactions occur with plastics through the enzymes produced, breaking chemical bonds between plastic molecules or polymers	* Alternaria alternate * * Aspergillus tubingensis * * Penicillium oxalicum * * Pestalotiopsis microspora *	[107]
Pesticide-tolerant fungi	Organochlorine, phosphorus and other pesticides	Overused farmland	Secretory enzymes form a degradation system, decomposes pesticide molecules into inorganic compounds through co-metabolism and mineralization	* Phanerochaete * * Acremonium * sp. * Rhodococcus * sp. * Paecilomyces lilacinus *	[108,109]
basidiomycetes	polycyclic aromatic hydrocarbons, polychlorinated biphenyls	Soil Water	intracellular segregation, organic acid precipitation and metal binding proteins	*Aspergillus niger* *Podospora anserina*	[110]
Mushroom	Heavy metal, Dyes, etc.	Wastewater and field	Mycelium cell wall has ion adsorption capacity	*Trachyderma lucidum* *Phoenix mushroom*	[111,112]
Algae	water contamination	Water	Photosynthesis absorbs carbon dioxide and nutrients such as nitrogen, phosphorus and ammonia in wastewater. Both adsorption and desorption mechanisms are more effective in living cells than in dead cells	* Phycoremediation *	[113]

Biofilms have specialized pollutant degradation functions, enabling the release and diffusion of functional genes (as single/double-stranded DNA or circular plasmids) [114]. These genetic materials enter new hosts through gene recombination and conjugation, facilitating horizontal gene transfer across same or different microbial species, effectively enhance the biofilm’s ability to remediate contaminated soil. Consequently, some studies aim to improve microorganisms’ pollutant degrading metabolic capacity through gene editing, aiming to apply these engineered organisms in agricultural settings [115].

However, engineered bacteria and fungi have limited natural survival, and the repair cycle remains lengthy. Efforts to modify their competitive dynamics with native microorganisms via antisense RNA and suicide genes have had minimal impact. Moreover, uncontrollable environmental factors, as well as pollutants (which may have synergistic effects and concealment), further complicate bioremediation. Beyond conventional inorganic heavy metals and microplastics, new persistent organic pollutants (e.g., endocrine disruptors, antibiotics) have emerged. Both single-species and colony biofilms are primarily effective in remediating similar pollutants in contaminated soils; however, research on the remediation of composite pollutants in diverse ecosystems remains limited [116].

Compared to animal and plant polysaccharides, microbial exopolysaccharides have drawbacks (instability, vulnerability to crop loss, soil pollution, climate change) but also advantages (low toxicity, biodegradability, sustainable use potential). Functional biofilms developed through Biofilm Integrated Nanofiber Display (BIND) technology have enhanced bioactive. Nguyen et al. engineered *Escherichia coli* biofilms to express amyloid CsgA fusion protein within the extracellular matrix, facilitating self-assembly into an extracellular amyloid nanofiber network [117]. This innovation improves biofilm stability and substrate adhesion, effectively transforming it into tailorable biological repair material. Subsequently, it is essential to clarify EPS functional groups’ characteristics and classification, and explore strategies (genetic manipulation, synthetic biology, etc.) to boost EPS yield in modified strains, such as *A. niger* has been widely used as cell factory for polysaccharide production [118]. It is important to examine the functions and synthesis mechanisms of multifunctional biofilms, as well as the combined repair capabilities of multiple strains.

Furthermore, integrating multi-omics approaches, such as metagenomics, with systems biology methodologies is essential for achieving a comprehensive understanding of the commonalities across various terrestrial and aquatic ecosystems. This integration will facilitate the development of programmable biofilm materials or artificially optimized biofilm communities that exhibit a broad degradation spectrum, high repair efficiency, and robust repair capabilities tailored to specific requirements. Ultimately, these advancements aim to enhance the restorative capacity of biofilms within ecological environments.

### 5.2. Host Growth Promotion

In agricultural and horticultural systems, rhizosphere microorganisms are widely deployed as biostimulants to foster crop growth and boost yields. Hormone analogs synthesized by these microorganisms mimic endogenous plant hormones, influencing not only on local plant tissues but also indirectly modulating the aboveground biomass and overall performance of host plants [119,120]. This regulatory impact is mediated by altering traits such as root system architecture and mineral nutrient acquisition, thereby achieving long-term impact across various spatial scales. For instance, research conducted by Shen Qirong’s team demonstrated that volatile organic compounds (VOCs) produced by beneficial rhizosphere bacteria can enhance the productivity of individual plant communities by stimulating phenotypic variation, resulting in a pronounced overproduction effect [121].

Plants and microorganisms have coevolved to establish highly mutualistic symbiotic relationship. Rhizosphere microorganisms serve as indicators of plant health, as their community composition and metabolite profiles accurately reflect the physiological status of host plants, which can be utilized to develop targeted plant health management strategies. Monitoring rhizosphere microbial dynamics enables the early detection of plant diseases [122,123]. Conversely, the application of beneficial microbial activation signals can indirectly fortify crop defense mechanisms and reduce reliance on chemical pesticides. In transgenic crop research, efforts are focused not only on modifying genes that directly influence crop growth indices but also on introducing genes that mediate the assembly of beneficial rhizosphere microbial communities, thereby optimizing the microenvironment to indirectly promote crop growth [124]. Bacteria are particularly amenable to such biotechnological development due to their unicellular nature. For instance, bioselenium nanoparticles (SeNPs) synthesized by selenium bacteria can induce chemotaxis and biofilm formation in plant growth-promoting rhizobacteria (PGPRs) in a dose-dependent manner. This approach reveals a novel method for recruiting beneficial soil microorganisms and proposes innovative strategies for accurately regulating plant-associated microbial communities [125].

Additionally, fungi such as *Trichoderma* spp. are common host plant growth promoters [126]. As one of the three major microbial resources globally, due to survival advantages, strong adsorption cell structure, and ability to secrete extracellular enzymes (cellulase, chitinase, protease, phytase, etc.) and functional factors, it has been developed into pesticides, fertilizers, growth promoters, and soil remediation agents. Across *Trichoderma* species and strains, growth-promoting characteristics converge on synthesizing regulators/metabolites, modulating nutrient availability, regulating physiological processes, and enhancing stress resistance (see Table 5).

Regulatory effects of including modulating soil microbial diversity/enzyme activity, optimizing seedling biomass allocation, promoting tillering/earing, facilitating signal perception/transmission, inducing chlorophyll/soluble protein and adjusting K/Na ratio (*T. longibrachiatum* HL167 [141]); activating hormone signaling for defense (*T. longibrachiatum* H9 [142]); enhancing nutrient absorption with heat resistance (*T. longibrachiatum* MD30); producing growth-promoting volatile organics (*T. koningi* T-51 [140]); and regulating HCN while stimulating ammonia (*T. yunnanense* TM10 [143]). Some (*T. harzianum* T22 [131]) also accelerate germination, boost seedling vitality, and induce oxidative damage protection.

### 5.3. Enhanced Host Resistance

Throughout long-term evolution, plants have developed a variety of defense mechanisms that are activated in response to environmental stressors. These mechanisms include enhanced physical barriers, such as the cuticle structure, as well as the secretion of fungicides and other bioactive compounds, collectively referred to as systemic acquired resistance (SAR) [144]. This adaptive response enhances the plant’s defense against pathogens, nematodes, and parasitic plants, enabling localized disease control during the initial stages of infection. Upon subsequent infection by the same pathogen, the plant’s SAR can facilitate a more rapid and effective resistance response. Microorganisms can bolster plant stress resistance, particularly through the colonization of the rhizosphere by specific beneficial microbial species, which can trigger induced systemic resistance (ISR) in plants [144]. This ISR enhances the resistance of neighboring plants without directly activating their own defense responses, which is influenced by competition among microorganisms [145].

Exploitation competition occurs when microorganisms consume limited resources and rapidly occupy spatial niches, thereby forming a physical barrier that effectively excludes pathogens. Rhizosphere microorganisms can chelate insoluble iron oxides or iron hydroxides by secreting ferriferous carriers, which allows them to quickly utilize iron chelates through specific extracellular receptors, thereby limiting the availability of essential nutrients to competing microorganisms and indirectly diminishing their survival capabilities [146]. Conversely, interference competition involves antagonistic interactions among microorganisms, wherein they secrete effectors to inhibit the growth of competitors. Antagonistic substances that can directly suppress pathogen growth include bacteriocins, antibiotics, lysozymes, and proteases [147]. Additionally, interference signals can indirectly inhibit other microorganisms by disrupting cell communication through mechanisms such as quorum quenching (QQ) [148]. This disruption impedes quorum sensing, preventing pathogens from accurately perceiving population density and hindering their ability to form biofilms, thereby weakening their pathogenicity. These interference signals are known as quorum-sensing inhibitors (QSIs), which include VOCs such as ethylene, jasmonic acid (JA), and salicylic acid (SA), as well as fatty acids (FAs) and their derivatives [149].

The application of rhizosphere microbial-induced systemic resistance offers opportunities for the development of safer and more environmentally sustainable biological strategies aimed at enhancing plant quality and combating plant diseases. Research has demonstrated that the EPS produced by *Bacillus thuringiensis* can limit the spread of *Sclerotinia sclerotiorum* in rapeseed through ISR [150]. Qian et al. revealed that rhizosphere bacteria inhibit the quorum-sensing system by blocking the synthesis of the quorum-sensing molecule AHL in pathogenic bacteria [151]. This finding represents the first documented instance of QQ that does not involve the degradation of QSMs but rather interferes with their synthesis. Directly applying artificially optimized microbial strains to crop cultivation soil or utilizing fermentation products in the plant rhizosphere can help reduce soil biological barriers and improve crop yields. Research has shown that the application of the proteolytic *Bacillus* OSUB18 as a root infusion can activate ISR in *Arabidopsis*, enhancing the plant’s resistance to *Pseudomonas syringae* and *Botrytis cinerea* [152]. Currently, a summary of commonly utilized biocontrol microorganisms is presented in Table 6, which highlights the five most prevalent microbial pesticide varieties: *Bacillus thuringiensis*, *Bacillus subtilis*, *Heliothis armigera nucleopolyhedrovirus* (HaNPV), *Metarhizium anisopliae* CQMa421, and *Bacillus polymyxa* KN-03.

Furthermore, research has demonstrated that the combination of multiple bacterial or fungal strains often yields more significant effects than the application of a single strain, including unexpected outcomes that cannot be achieved by any individual strain. For instance, the ARC bactericide developed by Li’s research team consists of four distinct bacteria: *Bacillus amyloliquefaciens*, *Brevibacillus laterosporus*, *Bacillus mucilaginosus*, and *Enterobacter ludwigii*. Remarkably, it shows substantial benefits in improving quality, fixing nitrogen, and inhibiting *A. flavus* only when all four strains coexist. This phenomenon is believed to be linked to specific signaling molecules generated through their interactions; however, the precise mechanisms underlying this effect remain to be elucidated [153]. Currently, ARC is widely used in peanut and soybean cultivation across various provinces in China.

**Table 6 jof-11-00726-t006:** Microbial agent products for pathogen control.

Genus	Species	Applied	Control	Product (Country)	References
*Bacillus*	*B. amyloliquefaciens* *B. cepacia* *B. cereus* *B. laterosporus* *B. lichenifomis* *B. subtilis* *B. thuringiensis* *B. velezensis*	Various grain Cotton Vegetable Fruit Arabidopsis	Sheath blight Blight Chlorotic wilt Black rot, Anthrax, Rice blast Lepidoptera, Root-knot nematode	ARVATICO (China) Double nickel (Canada) XenTari WG (Canada) Companion (USA) Provilar (USA)	[154,155]
*Pseudomonas*	*P. fluorescens* *P. putida* *P. brassicacaerum* *P. oryzihabitans* *P. marginali*	Tomato, potato, cucumber, rape, strawberry, Cotton, Various grain, Tobacco	Bacterial wilt, Plague, Gray mold, Root rot, Black shank	GaoTian (China)	[156,157,158]
*Agrobacterium*	*A. amazonense* *A. radiobacter* *A. rhizoides* *A. larrymoorei*	Fruit tree, rice	Root cancerosis	K599 (China)	[159]
*Collimonas*	*C. fungivorans*	Tomato, cabbage, watermelon, jujube	Scab Black rot	Complex biological agent (China)	[160]
*Comamonas*	*C. acidovorans* *C. aquatica*	Rice Kiwi	Sheath blight	Ca30 (China)	[161]
*Trichoderma*	*T. harzianum* *T. longibrachiatum* *T. viride* *T. asperellum* *T. atroviride*	Various grain, vegetable, fruit. Agricultural and forestry fields	Dry rot, Wilt, Verticillium	Trianum (Canada) TrichoPlus (USA) RootShield Plus (USA) Trichodex^®^, Trianum^®^ (Holland)	[162,163]
*Streptomyces*	*S. jingyangensis* *S. bungoensis* *S. fimbriatus* *S.lydicus* *S.griseoviridis*	Vegetable, Pear	Anthrax, Gray mold Black blot	TJ561 (China) Actinovate^®^, Mycostop^®^	[164]
*Aspergillus*	*A. oryzae* *A. niger* *A. awamori* *A. glaucus* *A. terreus*	Rice Maize Peanut	Blast	CGMCCNo.7127 (China)	[165]
*Penicillum*	*P. frequentans* *P. baileii* *P. citri* *P.P.P.*	Cucumber Apple Maize Wheat Cotton Sesame Tobacco	Wilt Verticillium Anthracnose Root rot botrytis	MoonBiotech (China)	[166]
*Paecilomyces*	*P. lilacinus*	Various fruits and vegetables, tobacco, rice, oil crops	Plant pathogenic nematode	JBC033 (China)	[167]
*Beauveria*	*B. bassiana*	Forest, nurserie, lawn, field	Grub, ostrinia nubilalis Dendrolimus Empoasca vitis	Tackler (India) Tezpetix Beauve (USA) ARBIOGY (China)	[168,169]
*Metarhizium*	*M. anisopliae* *M. ablum* *M. cylindrosporum* *M. acridum* *M. lepidiotae* *M. globosum* *M.robertsii* *M.majus* *M.rileyi*	Agriculture and forestry, fruit trees, fields	Scarabean beetle Weevil Wireworm Lepidoptera Hemiptera	JULIXIN (China)	[170,171,172]

## 6. Discussion

The interactions between plant and microbial communities are more intricate than mere pairwise relationships, which constitute a dynamic, multi-layered network woven from metabolic crosstalk, signal transduction, and co-evolutionary feedback loops [63]. These interactions operate across spatial scales, from the microenvironments of root hairs to entire agroecosystems. Such complexity renders obsolete the reductionist approaches of the past, which isolated individual microorganisms or plant species for study. It is now imperative to adopt a systems ecology perspective, one that integrates metagenomic data, metabolomic profiling, and environmental monitoring to capture the full breadth of interactions. This shift in framework is not merely a methodological choice but a scientific necessity, yet it presents profound challenges, from the computational hurdles of modeling 5000+ interacting species in a single soil sample to the technical limitations of tracking transient quorum-sensing molecules like acyl-homoserine lactones in real time [173].

Nevertheless, the rhizosphere microbiome has demonstrated a remarkable resilient and effective tool in agricultural practice, even amid incomplete mechanistic understanding. The urgency to unlock this potential has intensified in recent years, driven by two global imperatives: the “double reduction” policy targeting pesticide and fertilizer usage (a response to widespread soil degradation, such as cadmium contamination in 16% of Chinese farmland [174], and eutrophication of 50% of U.S. lakes [175]) and the race to achieve carbon neutrality (which demands agricultural systems that sequester 0.5–1.0 Pg of carbon annually). Together, these goals have transformed the development of innovative biological pesticides and biofertilizers from a niche pursuit into a strategic priority. However, the industrialization of these microbial products faces a labyrinth of technical barriers including the short shelf life of microbial agents, inconsistent efficacy, poor adaptability, and slow action, which stymied market penetration.

To overcome these challenges, researchers are turning to biomimicry that is adhering to the principles of natural systems to design more robust, cost-efficient solutions. Advances in genetic engineering technology are taking this further, shifting focus to synthetic microbial communities (SMCs) [176], as well as products that exhibit long-term stability and extended shelf life [177]. Beyond product design, the future of microbial agriculture lies in holistic integration into crop management, for example, rather than one-time inputs, “full-cycle application” is gaining traction [178,179,180].

This evolution reflects more than technological progress; it represents a fundamental shift in humanity’s relationship with nature. For decades, agriculture sought to dominate ecosystems—now, we collaborate with them. The rhizosphere microbiome once hidden has become a cornerstone of this approach. As we deepen understanding and refine engineering, we move closer to a future where agriculture is not just productive, but regenerative: feeding 9.7 billion people by 2050 while healing degraded soils and mitigating climate change.

## Figures and Tables

**Figure 1 jof-11-00726-f001:**
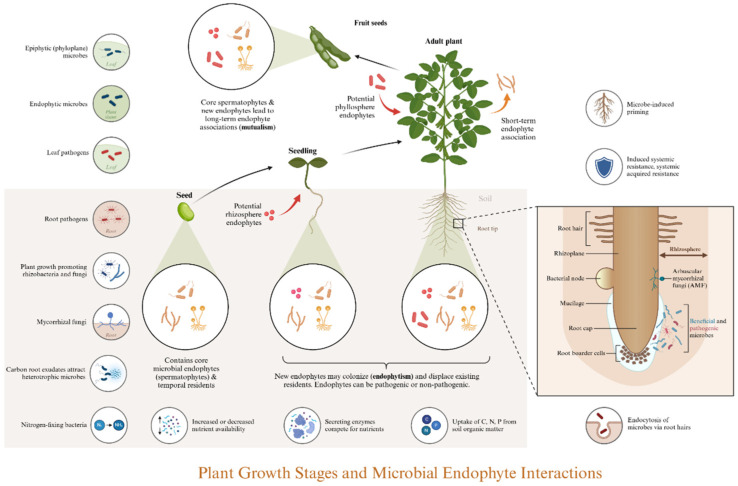
Dynamic Changes in the Rhizosphere Microbiome Associated with Plant Hosts. As plants mature, microorganisms are progressively recruited to the rhizosphere in the soil, resulting in a balanced rhizosphere microbiome with varying compositions at different developmental stages. This microbiome includes neutral symbiotic, beneficial growth-promoting, harmful pathogenic bacteria and fungi, all of which directly and indirectly influence the growth of plant hosts.

**Figure 2 jof-11-00726-f002:**
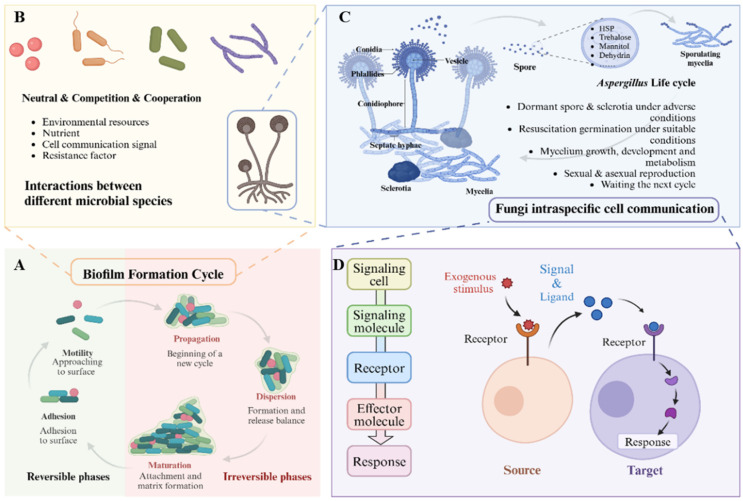
Interspecific interactions and intraspecific communication mechanisms of fungi within the rhizosphere microbiome. (**A**) Biofilm formed on the root surface exemplifies the interaction among various microorganisms. (**B**) Within these biofilms, relationships among members can be categorized as neutral, competitive, or cooperative, influenced by factors such as environmental resources, nutrient availability, cellular communication signals, and resistance mechanisms. (**C**) For example, in fungal biofilms, particularly those involving *Aspergillus*, the life cycle of individual strains is profoundly affected by cellular communication. This process encompasses the activation of dormant spores under unfavorable conditions; when conditions improve, these spores commence germination, followed by hyphal growth, development, and metabolic activity, ultimately culminating in the next reproductive cycle through either sexual or asexual means. (**D**) Intraspecific cells respond to external stimuli through ligand-receptor interactions. Essential components of this process include the exogenous stimulus, the perception and secretion of signals by source cells, the reception of these signals by target cells, and the subsequent activation of intracellular pathways in response.

**Table 1 jof-11-00726-t001:** Advanced methods in research of rhizosphere interactions.

Method Category	Key Techniques
DSM Identification	LC-MS/MS; UHPLC-Q-TOF-MS (untargeted profiling of novel terpenes); Ion mobility spectrometry-MS (IMS-MS); Surface-enhanced Raman spectroscopy (SERS, label-free detection of low-abundance molecular)
Microbial Cultivation	Microfluidic chips (for single-cell cultivation); SynComs (defined synthetic microbial signaling communities); Hypha-on-a-chip platforms (real-time imaging of molecular secretion)
Microbiome & Interaction Analysis	High-throughput ITS2 metabarcoding (with PacBio Sequel II for species-level fungal profiling); Confocal Raman microscopy (in situ mapping); Single-molecule fluorescence (in situ hybridization); NanoSIMS (trace nutrient exchange linked to DSM signaling)
Metabolomics & Omics Integration	Spatial metabolomics (map DSM distribution across microdomains); Meta-transcriptomics (Iso-Seq for full-length fungal transcripts); Multi-omics integration (LC-MS/MS + RNA-seq + 16S/ITS sequencing via tools like MetaboAnalyst)
Biosensing & Real-Time Detection	Whole-cell biosensors (engineered cells with GFP); Electrochemical biosensors (gold-nanoparticle modified electrodes); Fluorescent aptamers (targeted imaging of molecular in plant roots)
Computational & Systems Biology	Genome mining (antiSMASH + PRISM for predicting gene clusters); Metabolic flux (COBRA toolbox to simulate DSM production)

**Table 2 jof-11-00726-t002:** Fungi associated with plant diseases in agricultural ecosystems and their ecological–agronomic contexts.

Fungal Genus	Pathogenic Species	Infection Site	Cause Disease	Disease Characteristics	Affected Crop/Plant	References
Erysiphe	*E. cichoracearum* *E. polygoni*	Leaf Branch Flower	powdery mildew	Damage by toxin, white powder appears at the affected part and develops into blister points, resulting in fallen leaves, scorched tips, poor growth, etc.	Melons Beans Polygonaceae chenopodiaceae	[40]
Rhizoctonia	*R. leguminicola* *R. solani* *R. zeae*	Root Stem Leaf	Root, stem rot Seedling blight Stem canker Black spot	Toxins and degrading enzymes destroy cell walls and cause root rot	Grains Cotton Tobacco Beans Flowers	[41,42]
Pasasitica	*Hyaloperonospora parasitica* *Peronospora destructor* *Plasmopara viticola* *Pseudoperonospora cubensis* *P. tabacina*	Leaf Shoot	Downy mildew	Grayish-white frosty substance appears at the leaf back side and develops into polygonal lesion, characterized by swelling or deformity	Cruciferous vegetables Leguminosae Allium Grapes Calabash Tobacco	[43,44]
Meliolales	*M. butleri* *M. camellicola* *M. schimicola* *M. ardinis* *M. phyllestachydis*	Leaf	Fumigant mold Sooty disease Sooty blight	Leaf covered with coal dust produced by radiative mildew that inhibits plant photosynthesis, and toxins cause curl and fall off.	Citrus Theaceae Schima Poaceae Bamboos	[45]
Colletotrichum	*C. fructicola* *C. karsti* *C. siamense* *C. fioriniae* *C. gloeosporioides* *C. acutatum* *C. kingianum* *C. viniferum* *C. truncatum*	Leaf Fruit	anthracnose	Symptoms on petioles, tendrils, and young stems appear as light brown spots, which gradually change from round to oval concave necrotic spots. Form dark brown lesions and pink gelatinous mucus	Fruit trees Polygonatum Magnolia Grapes	[46,47]
Alternaria	*A. iridicola.* *A. tenuissima* *A. longipes* *A. alternata* *A. solani*	Leaf	Leaf spot Tobacco brown spot Tomato Early Blight	Leaves were locally infected with spots of various shapes and colors	Cereal crops oil crops vegetables and fruits ornamental flowers Chinese herbs Tobacco	[48]
Sclerotinia	*S. sclerotiorum* *S asari* *S. trifoliorum* *S. rolfsii*	Leaf Stem Root near the earth surface	White blight White rot Sclerotiniose	White silk hypha then develops to sclerotium, can produce virulence factors that inhibit plant immune system and even induce plant cell death	Mono-/Dicotyledon Leguminosae Tobacco Solanaceae Fruit trees	[49]
Phytophthora	*P. sojae* *P. capsici* *P. ramorum* *P. citrophthora* *P. meadii* *P. parasitica* *P. palmivora*	Petiole of leaf	Late blight	Secreting toxins that interfere with plant cell metabolism	Potatoes Tomatoes Soybeans Citrus Palm trees	[50,51]
Puccinia	*P. sorghi* *P. polysora* *P. coronata* *P. striiformis* *P. graminis*	Apex bud, lateral bud of young and large trees	rust	Secreted virulence factors can interfere with the metabolism of plant cells, resulting in rusty dry rot on the leaf edge	Wheat Corn Orchardgrass	[52]
Ascochyta	*A. graminicola* *A. hordei* *A. sorghi* *A. melongena*	Root Leaf	Vine blight Ring spot Brown spot	Toxins and degrading enzymes destroy the cell wall, and transfer to the whole plant through the tube system.	Elder trees Nightshade Cucurbitaceae	[53,54]
Colletotrichum	*C. gloeosporioides* *C. acutatum* *C. boninense* *C. orchidearum* *C. gigasporum*	Leaf	Rot disease Anthracnose	Pathogenic factors caused bark rot and xylem damage, inhibiting water and nutrient transport.	Litchi Kiwi Apple Cowpea Soybean	[55]
Fusarium	*F. graminearum* *F. avenaceum* *F. oxysporum*	Root Stem Spike	Blight Gibberellosis Leaf spot	Secreted toxins and degrading enzymes break down cell walls and cause root rot	Grains Fruit Tomato	[56,57]
Aspergillus	*A. flavus* *A. niger* *A. ochraceus* *A. parasiticus*	Root Fruit	Aspergillosis rot Mycotoxin contamination	Toxigenic	Grains Beans Coffee Onions Grapes	[58]
Cytospora	*C. chrsosperma* *C. pyriformis* *C. poplar* *C. willows*	Main branch Trunk Fruit	Rot disease	Infection causes the plant host to ulcerate or wilt	Willow Fruit tree Poplar Eucalyptus	[59]
Boeremia	*B.exigua*	Root	Leaf spot Root rot	Caused root vascular bundle necrosis	Red euphorbia	[60]

Note: The ‘pathogenic’ trait described herein is specific to agricultural ecosystems, where these fungi disrupt crop productivity. In natural ecosystems, many of these taxa (e.g., *Fusarium* spp.) function as decomposers or participate in plant–fungal symbioses, contributing to nutrient cycling and community regulation.

**Table 5 jof-11-00726-t005:** *Trichoderma* used to promote plant growth.

Species	Strains	Characteristic	Application	Reference
*T. asperellum*	HTTA-Z0002 CHF 78	Degrade the insoluble phosphorus and potassium, produce phytase, cellulase, indole-acetic acid and iron carriers that promote plant growth	Tomato	[127]
*T. harzianum*	ACCC31649	Produce plant growth regulators that can regulate exogenous plant hormones bidirectionally	Pepper	[128]
	T22	Produce plant hormones that can accelerate seed germination, increase seedling vitality, and induce physiological protection of plants against oxidative damage	Tomato seeds, cherry wood	[129]
	T2-16	Regulating microbial diversity in soil, increasing enzyme activity, and promoting effect on plant	Watermelon	[130]
	ESALQ 1306	Reduce the roots biomass allocated by seedlings, which can promote seedling growth	Wheat	[131]
	IMI 206040	Produce hormones that stimulate plant growth and development	Arabidopsis thaliana	[132]
	OTPB3	Stimulate the secretion of hormone and induce plant defense system	Tomato	[133]
	IRRI-2,3,4,5,6	Promote tillering and earing of rice, increase the total number of grains per ear, and increase fertility rate	Rice	[134]
*T. viride*	KKP 792 DSM 1963	Produce ferric support, increases the activity of 1-aminocyclopropane-1-carboxylic acid deaminase (ACCD), and has the effect of phosphate solubilization	Rape	[135]
	TR-7	Produce indole acetic acid, ferric support, and extracellular enzymes that promote growth by colonizing root hairs	Cabbage	[136]
	Gv. 29-8	Produce auxin analogs, which promote root development	Arabidopsis thaliana	[137]
*T. koningi*	IM 0956	Stimulate the production of iron carrier, phosphate dissolution and ACC deaminase activity	Rape	[136]
	T-51	Produce volatile organic compounds that have growth-promoting effects	Arabidopsis thaliana	[138]
*T. longibrachiatum*	HL167	Induce the production of chlorophyll and soluble protein, and increase K/Na ratio, which can promote plants growth	Cowpea	[139]
	H9	Regulates defense networks by activating signaling pathways associated with the plant hormones	Cucumber	[140]
	T(SP)-20	Stimulate plant product indole-acetic acid, ferriferous carrier, phospholytic enzyme	Peanut	[141]
	MD30	High temperature resistance, promote crop absorption and utilization of soil nutrients, improve the growth rate of seedling	Cucumber	[141]
*T. yunnanense*	TM10	Stimulate the production of indole-acetic acid, phospholytic enzyme, ammonia, regulate HCN and other mechanisms to promote rice growth and germination.	Rice	[142]

## Data Availability

No new data were created or analyzed in this study.

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
