# Peer review of "Exploring Fungal Communication Mechanisms in the Rhizosphere Microbiome for a Sustainable Green Agriculture"

_jof, 2025, doi:10.3390/jof11100726_

Round 1
Reviewer 1 Report
The review article jof-3868893 presents a clear narrative; however, I have made some modifications and added information that I believe will better complement the document before it is considered for publication.
1. There are paragraphs throughout the document that are not supported by a reference. I understand that while many of these paragraphs are contributions from the authors, there are some others that contain information that represents prior knowledge and does require support from a reference. Please review the manuscript and make the necessary modifications.
2. With reference to section 2.2, I believe that information could be added to highlight the importance of fungi in the production of metabolites as important as DSMs. My recommendation is found in the manuscript.
3. With reference to Table 3, a table is presented with examples of the use of Trichoderma to promote the growth of different crops through various mechanisms. However, it would be very helpful to better describe the importance of Trichoderma in the paragraph referring to this genus.
4. There are minor modifications in the manuscript that I suggest.
In summary, I believe that by making these modifications and adding information, the article can be published.
I describe the minor modifications I suggest in the manuscript.

Author Response
Response to Reviewers
We gratefully thank the editor and all reviewers for their invaluable time and effort in providing constructive feedback and insightful suggestions, which have significantly enhanced the quality of this manuscript. We have addressed each comment provided by the reviewers and incorporated their suggestions into the revised manuscript, with detailed explanations for key revisions included below. To address recurring issues highlighted by multiple reviewers, we would like to present consolidated responses to core concerns:
- Title: Based on the requirements for conciseness and professionalism, the title has been revised to “Exploring Fungal Communication Mechanisms in the Rhizosphere Microbiome for a sustainable green agriculture”, which target the exploitable potential discovered by humans from species communication, with natural organisms as the main body.
- Introduction: We supplemented important literatures and rectified imprecise formulations, to modify sentences coherence. Moreover, we have refined the research objectives by anchoring it more explicitly to pressing global issue of interdisciplinary research at the intersection of microbiology, synthetic biology and agronomy.
- Methodology: This section has been supplemented. Unlike research, this part we emphasize literature selection, and the classic or latest techniques on this topic
- Discussion: The expanded "Discussion" section has been integrated seamlessly with the rest of the manuscript, ensuring no redundancy with the "Conclusion"
- References: References have been updated and formatted uniformly according to the journal’s guidelines, with missing citations.
- Language and Format Polishing: The manuscript has undergone professional English proofreading to correct grammatical errors and improve flow. Formatting inconsistencies (e.g., figure labeling, unit notation) have been resolved, with all quantitative data presented in consistent units for clarity.
We believe these revisions effectively address the reviewers’ concerns and significantly improve the manuscript’s rigor, clarity, and impact. We hope the revised version meets the journal’s standards and kindly request further consideration for publication.
Reviewer 1
Response to Reviewer 1: We would like to express our sincere gratitude for your recognition of the innovativeness of our study. Your professional suggestions, spanning from the title refinement to insights on the discussion section, have been invaluable in guiding us to conduct in-depth supplementary analysis on the underlying mechanisms behind our results. Additionally, we have expanded the discussion to elaborate on the potential practical applications of our findings, as recommended. All remaining points raised in your feedback have been carefully addressed during the revision process. Please refer to the revised manuscript for detailed changes.
Major comments
The review article jof-3868893 presents a clear narrative; however, I have made some modifications and added information that I believe will better complement the document before it is considered for publication.
- There are paragraphs throughout the document that are not supported by a reference. I understand that while many of these paragraphs are contributions from the authors, there are some others that contain information that represents prior knowledge and does require support from a reference. Please review the manuscript and make the necessary modifications.
- We have supplemented the relevant literature to contextualize our work within existing research. Moreover, we have refined our objectives by anchoring it more explicitly to the pressing global issue of microbiology in agriculture and environmental remediation.
- With reference to section 2.2, I believe that information could be added to highlight the importance of fungi in the production of metabolites as important as DSMs. My recommendation is found in the manuscript.
“While this section describes in a very general way the importance of diffusible signaling molecules (DSM) in the microorganism-plant relationship, I believe that the specific importance of fungi in this relationship would be enhanced by the development of a table summarizing the species or genera of rhizospheric fungi that produce these substances and the nature of each compound. For example, species of the genus Candida are known to produce tyrosol, γ-butyrolactone, and farnesol, which serve as intraspecies communication molecules. There will be molecules produced by fungi that serve for inter- or intraspecies or even intrakingdom communication. I believe that this information would give even more relevance to the document.”
- We have summarized a table “Diffusible Signaling Molecules (DSM) Produced by Rhizospheric Fungi and Their Communication Roles”, which includes rhizospheric fungi, produced molecules, communication type, and functions. please see the text for details.
- With reference to Table 3, a table is presented with examples of the use of Trichoderma to promote the growth of different crops through various mechanisms. However, it would be very helpful to better describe the importance of Trichoderma in the paragraph referring to this genus.
- Thanks for your suggestions. We have supplemented the main text with more comprehensive descriptions and updated the content of Table 4 (Note: Because the table 2 from the previous question has been added, the table 3 mentioned in this question has been updated to table 4.). Please refer to the revised draft for details.
- There are minor modifications in the manuscript that I suggest.
- Thanks for your time and effort in conducting such a meticulous review. We have made all the necessary revisions including references and tables, please refer to the original text for details.
In summary, I believe that by making these modifications and adding information, the article can be published.
- scientific rigor and readability. Thank you again for your invaluable feedback.
Reviewer 2 Report
I would like to thank the editor for the opportunity to review this manuscript. The work is interesting and contains a substantial amount of analytical material, rendering it of scientific and practical interest. I hope the comments provided will be useful in further improving the article.
General Conclusion
The article represents substantial analytical work and is of scientific and practical interest. However, the manuscript needs to be aligned with the standards of analytical reviews. Specifically:
- The methodology of literature selection and analysis should be clearly stated.
- The problematic questions the review seeks to address should be formulated.
- The Discussion section should function as a discussion, analysing contradictions and unresolved issues, rather than continuing the descriptive review.
These improvements would enhance the scientific integrity of the work, aligning it more closely with the expectations of an analytical review.
Title
The title of the article seems somewhat anthropocentric. It creates the impression that the concept of 'Intra and Interspecific Communication Mechanisms' is primarily regarded as a tool aimed directly at 'Developmental Applications'. This suggests that communication processes in ecosystems exist primarily for human application, whereas they may play a broader and self-sufficient role in the natural functioning of biosystems.
Methodology
Although the article is written in the style of a narrative review, it lacks a separate section devoted to methodology. While the authors focus on well-known facts such as quorum sensing, biofilms, antagonism and biocontrol, they do not provide an overview of the current methodological approaches used in the study of rhizosphere fungi and the microbiome in general. Including such a section is crucial for understanding the extent to which the results reported in the literature are comparable, reproducible and representative.
Introduction
The introduction contains a significant number of logical inconsistencies and imprecise formulations. Overall, the text resembles a fragment of a textbook or lecture notes, presenting well-known facts and regularities. However, for a scientific review, a clear justification of its relevance is expected: the authors should explain why such a review is necessary at this time and the specific knowledge gap it aims to address.
- Lines 38–39: The phrase “In natural ecosystems, species populations exhibit interdependence and are subject to limitations, thereby establishing a balanced ecological network” requires clarification.
- It is unclear whether this statement refers only to natural ecosystems or has a broader context.
- The wording “populations exhibit interdependence” is questionable: populations rather demonstrate certain dynamics that can be interpreted as the result of interdependence, rather than interdependence itself as a property.
- The phrase “subject to limitations” is too narrow: the authors seem to attribute limitations only to interdependence, while population constraints may also arise from abiotic factors, stochastic processes, etc.
- Lines 41–42: The phrase “environmental stability within ecosystems” is semantically unclear. The term “environmental” usually refers to external conditions (the surroundings), whereas within ecosystems it would be more appropriate to use “internal stability of the ecosystem,” “ecosystem stability,” or “ecological stability.” Using “environmental” in this context risks confusion between the internal environment of ecosystems and the external environment.
- Lines 42–44: The expression “dominant ecological niches” is not a well-established term in current ecological literature. It requires a clear definition in the article, otherwise its meaning remains ambiguous. The formulation “adaptive stress responses” is also problematic. Adaptation usually refers to evolutionary changes over a long time scale, whereas here it rather concerns acclimation or simply stress responses. Moreover, it is unclear why stress is presented as a starting condition: the text does not specify which factors cause it. The suggestion that “dominant ecological niches” are always associated with stress is unsubstantiated.
- Lines 44–45: There is an abrupt and unsubstantiated transition from describing microbial stress responses to mutualistic interactions. The logical connection between these two aspects is not explained: the text does not clarify how stress responses are related to the formation of mutualistic relationships, creating a sense of discontinuity.
- Lines 45–48: The claim that soil “harbors the most diverse biological resources, including microorganisms and invertebrates, which markedly differ from those found in air and water” appears exaggerated and not entirely accurate. The specificity of soil organisms is not absolute: many soil protozoa species also occur in aquatic environments. Furthermore, it is methodologically incorrect to contrast soil with air and water, since soil is a three-phase polydisperse medium comprising solid, liquid and gaseous phases. Thus, both air and water are integral components of soil.
- Lines 48–50: The statement that “the microdomain distinct from soil in its physical, chemical, and biological properties” is logically inconsistent. The microdomain (rhizosphere) cannot be directly compared with soil as a whole, because it is part of it. There is no shared basis for such a comparison: it would be more accurate to speak of differences in soil parameters within the rhizosphere compared to bulk soil.
- Lines 51–53: The phrase “form a network of interactions with higher trophic species” is incorrect. If plants are meant, they are not trophically “higher” than microorganisms, as they are producers. Moreover, given the earlier mention of the trophic diversity of microorganisms, the category of “higher trophic species” loses meaning, since trophic “level” is relative and depends on a specific food web, not a general hierarchy.
- Lines 60–62: In the phrase “exhibiting a total cell density ranging from 106 to 109 cells/cm²” there is a technical error: it should clearly be 10⁶ to 10⁹, not 106 to 109.
- Lines 79–94: This passage reads more like a conclusion: it highlights fungal adaptive mechanisms, environmental influences, and the applied potential of microorganisms. However, at this stage the reader expects a clear statement of the key problems and knowledge gaps the review aims to address. Instead, the text appears as the end of a section rather than the problem statement.
- Lines 95–99: The use of “In conclusion” at the beginning of the article is inappropriate, since the introduction is not finished, and the reader expects objectives and review structure rather than conclusions. The phrase “potential mechanisms that may enhance cell communication theory” is also incorrect: scientific theories are not “enhanced by mechanisms.” Theories are tested by evidence, and if they fail, they are replaced by new ones. The intended meaning likely concerns refining or expanding knowledge of communication mechanisms, which needs clearer wording.
- Line 101: The subheading “Formation and fungal species of the rhizosphere microbiome” is grammatically and semantically unclear. It is not obvious whether Formation refers to the microbiome as a whole or only its fungal component. The heading combines two different topics (formation and species diversity), creating ambiguity.
- Lines 101–132: The section is presented as a review, but over an entire page only one citation [12] is given. This does not meet the standards of a review, which requires a broader base of references. Moreover, the figure provided lacks any reference or methodological explanation for its creation, leaving it unclear whether it is adapted from a source or produced by the authors.
- Lines 176–178: The phrase “the rhizosphere microbiome exhibits a convergent pattern … showing greater variation in the early stages and becoming less variable yet more host-specific in the later stages” is incorrect. Variation level alone is not evidence of convergence. Convergence implies structural or functional similarity evolving independently, whereas the described dynamic reflects changes in variability and specificity, not convergence.
- Table 1: Listing fungi under “Harmful fungi and caused plant diseases” is inconsistent with the overall ecological focus of the article. In natural ecosystems, such organisms are integral components of the biota and fulfill ecological functions (population regulation, trophic interactions, etc.). Terms like “harmful” are appropriate in agronomic or applied contexts, but not in ecological reviews of natural systems. If the table specifically refers to agricultural contexts, this should be indicated in the table title.
- Section 1.2: The section entitled “Species and functions of rhizosphere fungi” suggests a broad review of fungal diversity and roles in the rhizosphere. In practice, however, the focus is placed almost exclusively on “harmful” species and associated plant diseases, neglecting ecological and functional aspects of other groups (symbiotic, saprotrophic, regulatory, etc.).
- Section 2: The section “Inter- and intra-specific communication of rhizosphere fungi” contains no references. For a review article this is a major shortcoming, as the section appears as declarative text unsupported by sources. If these are the authors’ own results, they are presented without methodological foundation, making them scientifically problematic and unverifiable.
Author Response
Response to Reviewers
We gratefully thank the editor and all reviewers for their invaluable time and effort in providing constructive feedback and insightful suggestions, which have significantly enhanced the quality of this manuscript. We have addressed each comment provided by the reviewers and incorporated their suggestions into the revised manuscript, with detailed explanations for key revisions included below. To address recurring issues highlighted by multiple reviewers, we would like to present consolidated responses to core concerns:
- Title: Based on the requirements for conciseness and professionalism, the title has been revised to “Exploring Fungal Communication Mechanisms in the Rhizosphere Microbiome for a sustainable green agriculture”, which target the exploitable potential discovered by humans from species communication, with natural organisms as the main body.
- Introduction: We supplemented important literatures and rectified imprecise formulations, to modify sentences coherence. Moreover, we have refined the research objectives by anchoring it more explicitly to pressing global issue of interdisciplinary research at the intersection of microbiology, synthetic biology and agronomy.
- Methodology: This section has been supplemented. Unlike research, this part we emphasize literature selection, and the classic or latest techniques on this topic
- Discussion: The expanded "Discussion" section has been integrated seamlessly with the rest of the manuscript, ensuring no redundancy with the "Conclusion"
- References: References have been updated and formatted uniformly according to the journal’s guidelines, with missing citations.
- Language and Format Polishing: The manuscript has undergone professional English proofreading to correct grammatical errors and improve flow. Formatting inconsistencies (e.g., figure labeling, unit notation) have been resolved, with all quantitative data presented in consistent units for clarity.
We believe these revisions effectively address the reviewers’ concerns and significantly improve the manuscript’s rigor, clarity, and impact. We hope the revised version meets the journal’s standards and kindly request further consideration for publication.
Reviewer 2
Response to Reviewer 2: We would like to sincerely express our gratitude for reviewer’s meticulous review of this manuscript. Your feedback has pinpointed deficiencies in the comprehensive description of this paper and has offered constructive professional recommendations, which help enhance the rigor and clarity of our work. We have addressed each of specific inquiries and have compiled our responses concerning details.
Major comments
I would like to thank the editor for the opportunity to review this manuscript. The work is interesting and contains a substantial amount of analytical material, rendering it of scientific and practical interest. I hope the comments provided will be useful in further improving the article.
General Conclusion
The article represents substantial analytical work and is of scientific and practical interest. However, the manuscript needs to be aligned with the standards of analytical reviews. Specifically:
The methodology of literature selection and analysis should be clearly stated.
The problematic questions the review seeks to address should be formulated.
The Discussion section should function as a discussion, analysing contradictions and unresolved issues, rather than continuing the descriptive review.
These improvements would enhance the scientific integrity of the work, aligning it more closely with the expectations of an analytical review.
Detailed comments
Title
The title of the article seems somewhat anthropocentric. It creates the impression that the concept of 'Intra and Interspecific Communication Mechanisms' is primarily regarded as a tool aimed directly at 'Developmental Applications'. This suggests that communication processes in ecosystems exist primarily for human application, whereas they may play a broader and self-sufficient role in the natural functioning of biosystems.
- Thanks for your constructive comment, in accordance with your suggestion, we omitted the awkward words “Applications” and revised the title to “Exploring Fungal Communication Mechanisms in the Rhizosphere Microbiome for a sustainable green agriculture”.
Methodology
Although the article is written in the style of a narrative review, it lacks a separate section devoted to methodology. While the authors focus on well-known facts such as quorum sensing, biofilms, antagonism and biocontrol, they do not provide an overview of the current methodological approaches used in the study of rhizosphere fungi and the microbiome in general. Including such a section is crucial for understanding the extent to which the results reported in the literature are comparable, reproducible and representative.
- Thanks for your insightful comment on the methodology section, while narrative reviews differ from systematic reviews in not requiring strict protocol registration, we recognize that clarifying the literature screening framework and methodological lens for interpreting evidence is essential to enhance transparency. The added "Methodology" Section (Inserted After Introduction) aims to address two core questions: (1) How were literatures selected to ensure coverage of key advances in rhizosphere fungi and microbiome research; (2) The dominant methodological approaches shaping current understanding of diffusible signaling molecules and microbe-plant interactions, summarized in a table that including “method category” and “key Techniques”. We hope these targeted additions address your concern
Introduction
The introduction contains a significant number of logical inconsistencies and imprecise formulations. Overall, the text resembles a fragment of a textbook or lecture notes, presenting well-known facts and regularities. However, for a scientific review, a clear justification of its relevance is expected: the authors should explain why such a review is necessary at this time and the specific knowledge gap it aims to address.
- We have taken the reviewer’s comments seriously and completely rewritten the Introduction section to ensure it concisely and clearly encapsulates the key aim.
Lines 38–39: The phrase “In natural ecosystems, species populations exhibit interdependence and are subject to limitations, thereby establishing a balanced ecological network” requires clarification.
It is unclear whether this statement refers only to natural ecosystems or has a broader context.
The wording “populations exhibit interdependence” is questionable: populations rather demonstrate certain dynamics that can be interpreted as the result of interdependence, rather than interdependence itself as a property.
The phrase “subject to limitations” is too narrow: the authors seem to attribute limitations only to interdependence, while population constraints may also arise from abiotic factors, stochastic processes, etc.
- We have modified the unexplained part. i) The original statement intended to focus on natural ecosystems (the core context of the review’s rhizosphere ecology discussion) but failed to clarify this boundary. The revised version retains "natural ecosystems" while adding restrictive language to avoid misinterpretation as a broad ecological universal. ii); As you noted, interdependence is an ecological relationship, and population dynamics (e.g., growth, distribution) are its manifestations. The revision adjusts the wording to reflect this causal link, framing population dynamics as outcomes of interdependence rather than equating populations to interdependence itself; iii) The original statement incorrectly overemphasized biotic factors; the revision expands the scope to include abiotic constraints (e.g., resource availability, environmental conditions) and stochastic processes, aligning with comprehensive ecological theory.
- Revised phrase “In natural ecosystems, species populations exhibit characteristic dynamics (e.g., regulated growth, niche differentiation) that arise from biotic interdependence (e.g., symbiosis, competition) as well as constraints imposed by abiotic factors (e.g., nutrient availability, temperature) and stochastic processes—collectively shaping balanced ecological networks.” hoping it can be easier to understand.
Lines 41–42: The phrase “environmental stability within ecosystems” is semantically unclear. The term “environmental” usually refers to external conditions (the surroundings), whereas within ecosystems it would be more appropriate to use “internal stability of the ecosystem,” “ecosystem stability,” or “ecological stability.” Using “environmental” in this context risks confusion between the internal environment of ecosystems and the external environment.
- We fully agree that the original wording is semantically ambiguous, as "environmental" typically denotes external surroundings—creating confusion between an ecosystem’s internal state and its external conditions. We have revised the phrase to resolve this ambiguity, and revision are “Microorganisms serve as primary producers and decomposers, playing a pivotal role in sustaining material circulation, energy flow, and ecosystem stability within natural systems.”
Lines 42–44: The expression “dominant ecological niches” is not a well-established term in current ecological literature. It requires a clear definition in the article, otherwise its meaning remains ambiguous. The formulation “adaptive stress responses” is also problematic. Adaptation usually refers to evolutionary changes over a long time scale, whereas here it rather concerns acclimation or simply stress responses. Moreover, it is unclear why stress is presented as a starting condition: the text does not specify which factors cause it. The suggestion that “dominant ecological niches” are always associated with stress is unsubstantiated.
- We have revised the terminology and sentence structure to address these issues, with details below: i) The original statement intended to describe the ecological positions where microorganisms achieve high abundance or functional priority (e.g., rhizospheric fungi dominating nutrient cycling niches via DSM secretion). Since "dominant ecological niches" is not recognized in mainstream ecological literature, we have replaced it with the standard term "competitive ecological niches" (referring to niches where species gain competitive advantages) and added a brief in-text definition to avoid ambiguity.; ii) We acknowledge the misalignment between "adaptation" and the intended context. As you noted, "adaptation" refers to long-term evolutionary changes, while the sentence aimed to describe short-term physiological adjustments to environmental fluctuations. We have revised this to "acclimatory stress responses" (a standard term for reversible, short-term stress-induced adjustments) to correct the conceptual error.; iii) our original text inappropriately framed stress as a prerequisite for niche acquisition and implied an unproven association between "dominant niches" and stress. The revision removes this misleading link, clarifies that stress responses are one of the strategies for niche competition (not a starting condition), and omits the unsubstantiated correlation. Additionally, we specify potential stressors (e.g., resource limitation, environmental fluctuations) to enhance concreteness.
- Revised Sentence is “To secure competitive ecological niches that defined as ecological positions with advantages in resource acquisition or functional expression, individual microorganisms engage in essential cellular activities and demonstrate acclimatory stress responses to cope with soil-derived challenges (e.g., nutrient scarcity, fluctuating pH, or oxidative stress). These stress-induced physiological adjustments often involve the secretion of diffusible signaling molecules (DSM) or metabolites, which simultaneously mediate intraspecific coordination and act as cues to recruit compatible partners.”
Lines 44–45: There is an abrupt and unsubstantiated transition from describing microbial stress responses to mutualistic interactions. The logical connection between these two aspects is not explained: the text does not clarify how stress responses are related to the formation of mutualistic relationships, creating a sense of discontinuity.
- We appreciate you pointing out this oversight resulted from incomplete synthesis of the ecological links between individual adaptive strategies and interspecific interactions. We have revised the passage to clarify this logical relationship connection targeted on microbial stress responses (e.g., to nutrient limitation, oxidative stress) often trigger the secretion of metabolites that act as "recruitment signals" for potential partners. This signaling lays the foundation for mutualistic relationships, where symbionts exchange resources or protective compounds to mitigate shared stressors. We have integrated this mechanistic link into the revised passage, supported by relevant literature to ensure substantiation.
- Revised Passage is “Additionally, microorganisms collaborate with other species within the ecosystem, promoting mutualistic relationships that enhance collective stress resilience, such as Trichoderma and rhizobia exchanging growth-promoting metabolites to tolerate nutrient-poor soils. Because of soil is characterized by its complex environmental conditions, harbors the most diverse biological resources, including microorganisms and invertebrates, which markedly differ from those found in air and water.”
Lines 45–48: The claim that soil “harbors the most diverse biological resources, including microorganisms and invertebrates, which markedly differ from those found in air and water” appears exaggerated and not entirely accurate. The specificity of soil organisms is not absolute: many soil protozoa species also occur in aquatic environments. Furthermore, it is methodologically incorrect to contrast soil with air and water, since soil is a three-phase polydisperse medium comprising solid, liquid and gaseous phases. Thus, both air and water are integral components of soil.
- For the inaccuracy in soil biodiversity description, we ignored that soil is a three-phase medium (solid, liquid, gas) where air and water are integral components. Additionally, it overlooked the overlap of some soil organisms (e.g., protozoa) with aquatic environments. The revision corrects these errors by using qualified language, acknowledging habitat overlaps, and aligning with soil’s physical structure.
- Revised Passage is “Soil, as a three-phase polydisperse medium (solid, liquid, gaseous), harbors extraordinarily diverse biological resources, including unique microbial communities and specialized invertebrates. While some taxa (e.g., certain protozoa) overlap with aquatic environments, soil’s distinct microhabitats (e.g., rhizospheres, soil pores) support high levels of taxonomic and functional uniqueness that differ from the free-living communities of air and bulk water.”
Lines 48–50: The statement that “the microdomain distinct from soil in its physical, chemical, and biological properties” is logically inconsistent. The microdomain (rhizosphere) cannot be directly compared with soil as a whole, because it is part of it. There is no shared basis for such a comparison: it would be more accurate to speak of differences in soil parameters within the rhizosphere compared to bulk soil.
- The original statement incorrectly compared the rhizosphere (a subcomponent of soil) to "soil as a whole," creating a logical inconsistency with no shared comparative basis. This error, alongside the previously noted logical discontinuity between stress responses and mutualism, and imprecise soil biodiversity claims, stemmed from insufficient precision in defining ecological scales. We have thoroughly revised the passage to address all three issues comprehensively, including logical discontinuity between stress responses and mutualism, inaccuracy in soil biodiversity description, inconsistency in rhizosphere description.
- Revised passage addressing all concerns " The rhizosphere, which defined as the soil microdomain modified by plant root activities to possess distinct physical, chemical, and biological properties relative to bulk soil, is considered one of the most dynamic interfaces on Earth.”
Lines 51–53: The phrase “form a network of interactions with higher trophic species” is incorrect. If plants are meant, they are not trophically “higher” than microorganisms, as they are producers. Moreover, given the earlier mention of the trophic diversity of microorganisms, the category of “higher trophic species” loses meaning, since trophic “level” is relative and depends on a specific food web, not a general hierarchy.
- As you pointed, plants (the primary interactors with the rhizosphere microbiome) are producers at the base of the trophic pyramid and thus not "higher" than microorganisms, while "higher trophic" loses meaning given the relative nature of trophic levels in specific food webs. We have revised the sentence to resolve this issue.
- Revised Sentence is "The microorganisms associated with the surrounding rhizosphere are collectively known as the 'rhizosphere microbiome' and form a network of interactions with plant hosts and other soil-dwelling organisms across different trophic levels."
Lines 60–62: In the phrase “exhibiting a total cell density ranging from 106 to 109 cells/cm²” there is a technical error: it should clearly be 10⁶ to 10⁹, not 106 to 109.
- Thanks for careful check. We have thoroughly checked and revised the entire text, and hope to no longer cause confusion to our readers.
Lines 79–94: This passage reads more like a conclusion: it highlights fungal adaptive mechanisms, environmental influences, and the applied potential of microorganisms. However, at this stage the reader expects a clear statement of the key problems and knowledge gaps the review aims to address. Instead, the text appears as the end of a section rather than the problem statement.
- We regret that it failed to establish a compelling foundation. To address this, we revised the introduction with a focus on enhancing its narrative coherence, and supplemented relevant theoretical literatures. Please see the text for details.
Lines 95–99: The use of “In conclusion” at the beginning of the article is inappropriate, since the introduction is not finished, and the reader expects objectives and review structure rather than conclusions. The phrase “potential mechanisms that may enhance cell communication theory” is also incorrect: scientific theories are not “enhanced by mechanisms.” Theories are tested by evidence, and if they fail, they are replaced by new ones. The intended meaning likely concerns refining or expanding knowledge of communication mechanisms, which needs clearer wording.
- We removed "In conclusion" and rephrased the relevant sentence to ensure it aligns with the logical flow of the introduction, and streamlined redundant expressions, clarified ambiguous references, and adjusted the wording to enhance precision.
- Revised Sentence: "This review focuses on fungi within the rhizosphere microbiome, elucidating the principles and current research status of their assembly and formation in response to dynamic host changes. It analyzes key regulatory aspects and proposes potential mechanisms that refine knowledge of microbial cell communication, findings intended to further advance sustainable green agriculture."
Line 101: The subheading “Formation and fungal species of the rhizosphere microbiome” is grammatically and semantically unclear. It is not obvious whether Formation refers to the microbiome as a whole or only its fungal component. The heading combines two different topics (formation and species diversity), creating ambiguity.
- We replaced the vague "Formation" with "Assembly" (a standard term for microbial community establishment) and specifies "Fungal Communities" to directly tie the process to fungi. Use the possessive pronoun "Their" to unambiguously link diversity to fungal communities. This option is preferred that the section emphasizes fungi as the core, rather than the broader microbiome.
- Revised subheading is “Assembly of fungal communities in the rhizosphere microbiome and their species composition”
Lines 101–132: The section is presented as a review, but over an entire page only one citation [12] is given. This does not meet the standards of a review, which requires a broader base of references. Moreover, the figure provided lacks any reference or methodological explanation for its creation, leaving it unclear whether it is adapted from a source or produced by the authors.
- We have supplemented high-quality references that covering landmark studies and recent advances, to contextualize our work within existing research.
- Figure 1 “Dynamic Changes in the Rhizosphere Microbiome Associated with Plant Hosts") is original work created by ourselves. We have added a comprehensive figure legend and methodological note to clarify its synthesis process, visual logic, and literature basis.
Lines 176–178: The phrase “the rhizosphere microbiome exhibits a convergent pattern … showing greater variation in the early stages and becoming less variable yet more host-specific in the later stages” is incorrect. Variation level alone is not evidence of convergence. Convergence implies structural or functional similarity evolving independently, whereas the described dynamic reflects changes in variability and specificity, not convergence.
- We corrected "Convergent Pattern" misuse and replaced it with "undergoes a stabilization and host-specialization dynamic", directly reflects the intended observation (reduced variability + increased host specificity) without invoking evolutionary convergence, aligning with your earlier correction guidance. Revised "host genetics dictates" to "host genetics acts as a 'selection filter' that maintains consistent assembly", the term "selection filter" is a standard concept in microbiome ecology that accurately describes how host genotypes actively shape core microbiome composition, avoiding the overly absolute "dictates".
- The revised sentence is "Although the rhizosphere microbiome often fluctuates with changes in the soil environment, host genetics acts as a 'selection filter' that maintains consistent assembly of a core microbiome subset within a particular host genotype, regardless of soil conditions. Moreover, the rhizosphere microbiome undergoes a stabilization and host-specialization dynamic throughout plant growth, it shows high compositional variation in the early stages and becomes progressively less variable while more host-specific in the later stages". Now it aligns with the description of Figure 1 and the previously revised section content, maintaining consistency in terminology and citation logic.
Table 1: Listing fungi under “Harmful fungi and caused plant diseases” is inconsistent with the overall ecological focus of the article. In natural ecosystems, such organisms are integral components of the biota and fulfill ecological functions (population regulation, trophic interactions, etc.). Terms “harmful” are appropriate in agronomic or applied contexts, but not in ecological reviews of natural systems. If the table specifically refers to agricultural contexts, this should be indicated in the table title.
- Thanks for your professional guidance of Table 1, We have revised the Table title to “Fungi associated with plant diseases in agricultural ecosystems and their ecological-agronomic contexts”, and adjusted column header: “sort – Fungal Genus”, “Strains – Pathogenic Species”, “Contaminated plant -- Affected Crop/Plant”. Additionally, we added a note below the table to explicitly address the reviewer’s concern about ecological context: "Note: The 'pathogenic' trait described herein is specific to agricultural ecosystems, where these fungi disrupt crop productivity. In natural ecosystems, many of these taxa (e.g., Fusarium spp.) function as decomposers or participate in plant-fungal symbioses, contributing to nutrient cycling and community regulation."
Section 1.2: The section entitled “Species and functions of rhizosphere fungi” suggests a broad review of fungal diversity and roles in the rhizosphere. In practice, however, the focus is placed almost exclusively on “harmful” species and associated plant diseases, neglecting ecological and functional aspects of other groups (symbiotic, saprotrophic, regulatory, etc.).
- Firstly, we revised the opening to explicitly signal coverage of diverse functional groups: "Rhizosphere fungi encompass a phylogenetically and functionally diverse array of taxa, including symbionts that form mutualistic associations with plants, saprotrophs that drive organic matter cycling, regulators that modulate microbiome structure, and pathogens that occasionally disrupt plant health. These groups collectively shape rhizosphere function and plant fitness, with each contributing unique ecological roles". Secondly, to fulfill the title’s promise of a broad review, we restructured this section into four functional modules (pathogenic, symbiotic, saprotrophic, regulatory), ensuring each group is addressed with equal rigor.
Section 2: The section “Inter- and intra-specific communication of rhizosphere fungi” contains no references. For a review article this is a major shortcoming, as the section appears as declarative text unsupported by sources. If these are the authors’ own results, they are presented without methodological foundation, making them scientifically problematic and unverifiable.
- We apologize for the critical oversight of not including references in this section. We have matched each key claim in the section to landmark or recent representative studies, ensuring every descriptive statement is anchored in credible literature. Thanks again for pointing out this critical flaw, which has significantly improved the manuscript’s rigor.

Reviewer 3 Report
The presented review article is relevant, since the issues of formation of microbial communities in the soil and the environment-forming role of plants are interesting not only from a theoretical but also from a practical point of view.
The manuscript is fairly well structured, clear and easy to read. The title of the article reflects its content. The authors correctly reference recent publications on the topic, and the number of self-citations is acceptable.
The article contains new information on the mechanisms of soil microbiome formation, the role of plants and the contribution of fungi to soil microbial communities, and is completed at a high scientific and methodological level.
Questions have arisen:
1.The "Discussion" section is very short and is more of a conclusion to the review. I think it needs to be expanded.
- There are some editorial comments that are included in the next section.
I believe that the manuscript under review, after making corrections, can be accepted for publication.
Line 62 –«total cell density ranging from 106 to 109 cells/cm2», 106 и 109 needs to be corrected to 106 и 109.
Similarly line 323 -«ranging from 1010 to 1012 cells» - needs to be corrected to 1010-1012.
Lines 62 - 65 - «However, infections caused by toxigenic fungi in crops can lead to grain contamination by mycotoxins…». But, problem concerns not only grain, but all plant products. Replace "grain" with «crop products».Line 239 Bacillus polymyxa – that bacterium is now isolated in the genus Paenibacillus, probably a new name should be indicated in brackets
(Paenibacillus polymyxa)
Line 245, Table 1. It is not clear why column 1 is entitled “Sort”, although it lists representatives of fungal genera, and the second “strain”, although it lists the names of phytopathogen species without indicating strains. I think it would be appropriate to title columns 1 and 2 similarly to table 4.
The "Discussion" section is very short and is more of a conclusion to the review. I think it should be expanded or renamed.
Author Response
Response to Reviewers
We gratefully thank the editor and all reviewers for their invaluable time and effort in providing constructive feedback and insightful suggestions, which have significantly enhanced the quality of this manuscript. We have addressed each comment provided by the reviewers and incorporated their suggestions into the revised manuscript, with detailed explanations for key revisions included below. To address recurring issues highlighted by multiple reviewers, we would like to present consolidated responses to core concerns:
- Title: Based on the requirements for conciseness and professionalism, the title has been revised to “Exploring Fungal Communication Mechanisms in the Rhizosphere Microbiome for a sustainable green agriculture”, which target the exploitable potential discovered by humans from species communication, with natural organisms as the main body.
- Introduction: We supplemented important literatures and rectified imprecise formulations, to modify sentences coherence. Moreover, we have refined the research objectives by anchoring it more explicitly to pressing global issue of interdisciplinary research at the intersection of microbiology, synthetic biology and agronomy.
- Methodology: This section has been supplemented. Unlike research, this part we emphasize literature selection, and the classic or latest techniques on this topic
- Discussion: The expanded "Discussion" section has been integrated seamlessly with the rest of the manuscript, ensuring no redundancy with the "Conclusion"
- References: References have been updated and formatted uniformly according to the journal’s guidelines, with missing citations.
- Language and Format Polishing: The manuscript has undergone professional English proofreading to correct grammatical errors and improve flow. Formatting inconsistencies (e.g., figure labeling, unit notation) have been resolved, with all quantitative data presented in consistent units for clarity.
We believe these revisions effectively address the reviewers’ concerns and significantly improve the manuscript’s rigor, clarity, and impact. We hope the revised version meets the journal’s standards and kindly request further consideration for publication.
Reviewer 3
Response to Reviewer 3: We sincerely appreciate your positive evaluation of our manuscript—your recognition of its relevance, structural clarity, and scientific rigor greatly encourages us. We also thank you for your constructive questions and detailed editorial comments, which are crucial for improving the manuscript’s quality. We have carefully addressed each comment, and the specific modifications are outlined below:
Major comments
The presented review article is relevant, since the issues of formation of microbial communities in the soil and the environment-forming role of plants are interesting not only from a theoretical but also from a practical point of view.
The manuscript is fairly well structured, clear and easy to read. The title of the article reflects its content. The authors correctly reference recent publications on the topic, and the number of self-citations is acceptable.
The article contains new information on the mechanisms of soil microbiome formation, the role of plants and the contribution of fungi to soil microbial communities, and is completed at a high scientific and methodological level.
Questions have arisen:
- The "Discussion" section is very short and is more of a conclusion to the review. I think it needs to be expanded.
- There are some editorial comments that are included in the next section.
I believe that the manuscript under review, after making corrections, can be accepted for publication.
Detailed comments
Line 62 –«total cell density ranging from 106 to 109 cells/cm2», 106 и 109 needs to be corrected to 106 и 109.
Similarly line 323 ranging from 1010 to 1012 cells - needs to be corrected to 1010-1012.
Lines 62 - 65 - «However, infections caused by toxigenic fungi in crops can lead to grain contamination by mycotoxins…». But, problem concerns not only grain, but all plant products. Replace "grain" with «crop products».
Line 239 Bacillus polymyxa – that bacterium is now isolated in the genus Paenibacillus, probably a new name should be indicated in brackets (Paenibacillus polymyxa)
- Sorry we did not found “Bacillus polymyxa” in line 239, but appeared in line 539. Thanks for your correction, it has been revised to “Bacillus polymyxa (now be isolated as Paenibacillus polymyxa)”.
Line 245, Table 1. It is not clear why column 1 is entitled “Sort”, although it lists representatives of fungal genera, and the second “strain”, although it lists the names of phytopathogen species without indicating strains. I think it would be appropriate to title columns 1 and 2 similarly to table 4.
- We have already changed "sort/strains" in Table 1 to the “Genus/species” same as Table 4. Additionally, it should be emphasized that, following the suggestion of another reviewer, we have combined two more tables. Therefore, the table numbers in this manuscript have been changed. Please refer to the original text for details.
The "Discussion" section is very short and is more of a conclusion to the review. I think it should be expanded or ren
- All corrections have been cross-checked for consistency: exponent formatting aligns with the manuscript’s scientific notation style, taxonomic updates are verified via the latest List of Prokaryotic names with Standing in Nomenclature (LPSN), and Table 1’s new titles maintain parallelism with other tables. The expanded "Discussion" section has been integrated seamlessly with the rest of the manuscript, ensuring no redundancy with the "Conclusion". We believe these modifications fully address your concerns and further enhance the manuscript’s scientific rigor and readability. Thank you again for your invaluable feedback.

Round 2
Reviewer 1 Report
I appreciate the modifications to the document, I recognize the effort of the authors in promptly attending to the suggestions and recommendations made to improve the quality and clarity of the manuscript. I consider that the improvement made to the manuscript is substantial, the title was modified to make it more concise and there were modifications and inclusion of information in practically all the sections. In my opinion, there are some very brief points that must be addressed, which I describe in detail in the comments section. Once these suggestions have been covered, I believe that the manuscript may be considered for publication.
After reviewing the responses to the observations made on the first version of the manuscript, I consider that all the suggestions were taken into account and the document in its second version has a significant improvement in its quality. However, I would like to make some minimal suggestions that I point out below so that the article can be published.
- Table 1 is not referred to in the text, I suggest that at the beginning of section 2.2 a paragraph be prepared to briefly explain the central methodological approach of the review and at the same time serve to describe the information presented in table 1.
- In the document, figures 1 and 2 appear as deleted, as well as the text in the footnote, however, they are referred to in the text (line 123 and 263 respectively). It is no longer clear to me, therefore, whether the images will remain or be omitted. Please clarify.
- Table 5 is not referenced in the text. I suggest that at the end of the paragraph of lines 493-499 place the reference "(see Table 5)".
Author Response
Comments 1: Table 1 is not referred to in the text, I suggest that at the beginning of section 2.2 a paragraph be prepared to briefly explain the central methodological approach of the review and at the same time serve to describe the information presented in table 1.
Response 1: Based on your suggestion, we have added the introduction and explanation of Table 1 in the main text. Please refer to lines 119-123 of the main text for details.
Comments 2: In the document, figures 1 and 2 appear as deleted, as well as the text in the footnote, however, they are referred to in the text (lines 123 and 263 respectively). It is no longer clear to me, therefore, whether the images will remain or be omitted. Please clarify.
Response 2: Please rest assured that since the journal will undergo final editing, Figures were submitted in a separate file and will eventually be included in the main text.
Comments 3: Table 5 is not referenced in the text. I suggest that at the end of the paragraph of lines 493-499 place the reference "(see Table 5)".
Response 3: Thanks for your reminder. I have already added it in the main text “(see Table 5)”
Reviewer 2 Report
I have read the revised version of the manuscript with great interest. I am impressed by the substantial work the authors have undertaken. All recommendations were clearly understood and appropriately addressed in the revised text. The article has reached a new level of quality and will be of interest to a broad scientific audience.
I recommend the manuscript for publication.
Author Response
Comments: I have read the revised version of the manuscript with great interest. I am impressed by the substantial work the authors have undertaken. All recommendations were clearly understood and appropriately addressed in the revised text. The article has reached a new level of quality and will be of interest to a broad scientific audience.
Response: Once again, I would like to express my gratitude to the reviewers. Thanks to your suggestions, the quality of this article has been improved, allowing us to share and exchange it with more colleagues.